# Coresets for Relational Data and The Applications

**Jiaxiang Chen**[1]    **Qingyuan Yang**[2]    **Ruomin Huang**[1]    **Hu Ding**[*2]

[1]School of Data Science
[2]School of Computer Science and Technology
University of Science and Technology of China
{czar, yangqingyuan, hrm}@mail.ustc.edu.cn, huding@ustc.edu.cn

## Abstract

A coreset is a small set that can approximately preserve the structure of the original input data set. Therefore we can run our algorithm on a coreset so as to reduce the total computational complexity. Conventional coreset techniques assume that the input data set is available to process explicitly. However, this assumption may not hold in real-world scenarios. In this paper, we consider the problem of coresets construction over relational data. Namely, the data is decoupled into several relational tables, and it could be very expensive to directly materialize the data matrix by joining the tables. We propose a novel approach called "aggregation tree with pseudo-cube" that can build a coreset from bottom to up. Moreover, our approach can neatly circumvent several troublesome issues of relational learning problems [Khamis et al., PODS 2019]. Under some mild assumptions, we show that our coreset approach can be applied for the machine learning tasks, such as clustering, logistic regression and SVM.

## 1 Introduction

As the rapid development of information technologies, we are often confronted with large-scale data in many practical scenarios. To reduce the computational complexity, a number of data summarization techniques have been proposed [46]. **Coreset** is a well-known sampling technique for compressing large-scale data sets [43; 27]. Roughly speaking, for a given large data set $P$, the coreset approach is to construct a new (weighted) set $\tilde{P}$ that approximately preserves the structure of $P$, and meanwhile the size $|\tilde{P}|$ is much smaller than $|P|$. Therefore, if we run an existing algorithm on $\tilde{P}$ rather than $P$, the runtime can be significantly reduced and the computational quality (e.g., the optimization objectives for the machine learning tasks [53]) can be approximately guaranteed. In the past decades, the coreset techniques have been successfully applied for solving many optimization problems, such as $k$-means clustering [27; 20], logistic regression [33; 44], Gaussian mixture models [39], and continual learning [11].

Usually we assume that the input data is stored in a matrix such that the coreset algorithm can easily access the whole data. But this assumption may not hold in real-world scenarios. For example, according to the Kaggle 2017 "State of Data Science Survey", **65.5% of the data sets are relational data**. Namely, the data is decoupled into several relational tables, and we cannot obtain the data matrix (which is also called the "**design matrix**") unless joining all the tables. Relational database has a very large and fast growing market in this big data era. It is expected to reach USD 122.38 billion by 2027 [48]. Relational database has several favorable properties [54], e.g., the decoupled relational data can save a large amount of space, and it is friendly to several common statistical queries such as the aggregate functions COUNT, SUM and AVG in SQL. In recent years, the study on

---

[*]Corresponding author.

36th Conference on Neural Information Processing Systems (NeurIPS 2022).

"in-database learning" has gained a significant amount of attention in the areas of machine learning and data management [45; 35; 22; 37].

**However, constructing a coreset for relational database is much more challenging.** Suppose there are $s > 1$ relational tables and the size of the largest table is $N$. Let $n$ be the number of the tuples obtained by joining all the tables, and then $n$ can be as large as $O(N^s)$ [7] (we illustrate an instance of joining two tables in Table 1). Obviously it will take extremely large time and space complexities to join all the tables and train a model on such scale of data. To remedy this issue, a straightforward idea is trying to complete the computing task without explicitly constructing the design matrix. Unfortunately, even for some very simple computational tasks, their implementations on relational data can be NP-hard.

To see why these implementations are so hard, we can consider the following simple clustering assignment task (which is often used as a building block for coresets construction [27; 20]): suppose the $k$ cluster centers have already been given, and we want to determine the size of each cluster. Note that if we have the design matrix, this task is trivial (just assign each point to its nearest center, and then calculate the cluster sizes based on the assignment). However, if the data is decoupled into several relational tables, the problem of computing the assignment can be quite complicated. Khamis et al. [36] recently defined the problem of answering **F**unctional **A**ggregate **Q**ueries (FAQ) in which some of the input factors are defined by a collection of **A**dditive **I**nequalities[2] between variables (denote by FAQ-AI($m$), where $m$ is the number of additive inequalities). The problem of computing the clustering assignment can be formulated as an instance of FAQ-AI($k-1$) (to determine the cluster membership for one point (tuple), we need to certify that its distance to its own cluster center is lower than the distances to the other $k - 1$ centers). Recently, Khamis et al. [3] showed that evaluating even a FAQ-AI(1) instance is #P-hard, and approximating a FAQ-AI($m$) instance within any finite factor is NP-hard for any $m > 1$. Moseley et al. [42] also showed that even approximating the clustering assignment to any factor is NP-hard for $k \geq 3$. An intuitive understanding is that the query satisfying those inequalities require to access the information of each tuple on all the dimensions, where it is almost equivalent to constructing the entire design matrix.

## 1.1 Our Contributions

We consider *Empirical Risk Minimization (ERM)* problems in machine learning [56]. Let $\mathbb{R}^d$ be the data space. The training set $P = \{p_1, p_2, \cdots, p_n\} \subset \mathbb{R}^d$. But we assume that this set $P$ is not explicitly given, where it is decoupled into $s$ relational tables (the formal definitions are shown in Section 2). The goal is to learn the hypothesis $\theta$ (from the hypothesis space $\mathbb{H}$) so as to minimize the *empirical risk*

$$F(\theta) = \frac{1}{n} \sum_{i=1}^{n} f(\theta, p_i), \tag{1}$$

where $f(\cdot, \cdot)$ is the non-negative real-valued *loss function*.

Several coresets techniques on relational data have been studied very recently. Samadian et al. [49] showed that the simple uniform sampling yields a coreset for regularized loss minimization problems (note the uniform sampling can be efficiently implemented for relational database [59]). Their sample size is $\Theta(n^\kappa \cdot \mathtt{dim})$, where $\kappa \in (0, 1)$ (usually $\kappa$ is set to be $1/2$ in practice) and $\mathtt{dim}$ is the VC dimension of loss function (usually it is $\Theta(d)$). Thus the size can be $\Theta(\sqrt{n} \cdot d)$, which is too large especially for high dimensional data. Curtin et al. [22] constructed a coreset for $k$-means clustering by building a weighted grid among the input $s$ tables; the coreset yields a 9-approximation for the clustering objective. Independently, Ding et al. [24] also applied the "grid" idea to achieve a $(9 + \epsilon)$-approximation for $k$-means clustering with distributed dimensions (attributes). The major drawback of this "grid" idea is that the resulting coreset size can be as large as $k^s$ which is exponential in the number of tables $s$. Moseley et al. [42] recently proposed a coreset for $k$-means clustering on relational data by using the $k$-means++ initialization method [6], however, their approximation ratio is too high ($> 400$) for real-world applications. Moreover, the coresets techniques proposed in [22; 24; 42] can only handle $k$-means clustering, and it is unclear that whether they can be extended for other machine learning problems as the form of (1).

---

[2]Each additive inequality defines a constraint for the query; for example, in SVM, the "positive" class is defined by the inequality $\langle x, \omega \rangle > 0$, where $\omega$ is the normal vector of the separating hyperplane.

In this paper, we aim to develop an efficient and general coreset construction method for optimizing (1) on relational data. First, we observe that real-world data sets often have low intrinsic dimensions (e.g, nearby a low-dimensional manifold) [10]. Our coreset approach is inspired by the well-known Gonzalez's algorithm for $k$-center clustering [29]. The algorithm greedily selects $k$ points from the input point set, and the $k$ balls centered at these selected $k$ points with some appropriate radius can cover the whole input; if the input data has a low intrinsic dimension (e.g., the doubling dimension in Definition 3), the radius is no larger than an upper bound depending on $k$. Therefore, the set of cluster centers (each center has the weight equal to the corresponding cluster size) can serve as a coreset for the ERM problem (1), where the error yielded from the coreset is determined by the radius. Actually this $k$-center clustering based intuition has been used to construct the coresets for various applications before [34; 52; 21].

**However, we have to address two challenging issues for realizing this approach on relational data.** First, the greedy selection procedure for the $k$ cluster centers cannot be directly implemented for relational data. Second, it is hard to obtain the size of each cluster especially when the balls have overlap (the $k$ balls can partition the space into as many as $2^k$ parts). Actually, both of these two issues can be regarded as the instances of FAQ-AI($k-1$) which are NP-hard to compute [3; 42].

Our approach relies on a novel and easy-to-implement structure called "**aggregation tree with pseudo-cube**". We build the tree from bottom up, where each relational table represents a leaf. Informally speaking, each node consists of a set of $k$ cluster centers which is obtained by merging its two children; each center also associates with a "pseudo-cube" region in the space. Eventually, the root of the tree yields a qualified coreset for the implicit design matrix. The aggregation manner can help us to avoid building the high-complexity grid coreset as [22; 24]. Also, with the aid of "pseudo-cube", we can efficiently estimate the size of each cluster without tackling the troublesome FAQ-AI counting issue [3; 42].

Comparing with the previous coresets methods, our approach enjoys several significant advantages. For example, our approach can deal with more general applications. In fact, for most ERM problems in the form of (1) under some mild assumptions, we can construct their coresets by applying our approach. Also our coresets have much lower complexities. It is worth to emphasize that our coreset size is independent of the dimensionality $d$; instead, it only depends on the doubling dimension of the input data.

## 1.2   Other Related Works

The earliest research on relational data was proposed by Codd [18]. A fundamental topic that is closely related to machine learning on relational data is how to take uniform and independent samples from the full join results [13]. Recently, Zhao et al. [59] proposed a random walk approach for this problem with acyclic join; Chen and Yi [15] generalized the result to some specific cyclic join scenarios.

Beside the aforementioned works [45; 35; 22; 24; 36; 3; 42], there also exist a number of results on other machine learning problems over relational data. Khamis et al. [4] and Yang et al. [57] respectively considered training SVMs and SVMs with Gaussian kernel on relational data. For the problem of linear regression on relational data, it is common to use the factorization techniques [50; 2]. The algorithms for training Gaussian Mixture Models and Neural Networks on relational data were also studied recently [16; 17]. We also refer the reader to the survey paper [51] for more details on learning over relational data.

## 2   Preliminaries

Suppose the training set for the problem (1) is a set $P$ of $n$ points in $\mathbb{R}^d$, and it is decoupled into $s$ relational tables $\{T_1, \ldots, T_s\}$ with the feature (attribute) sets $\{D_1, \ldots, D_s\}$. Let $D = \bigcup_i D_i$ and therefore the size $|D| = d$. Actually, each table $T_i$ can be viewed as projection of $P$ onto a subspace spanned by $D_i$. To materialize the design matrix of $P$, a straightforward way is to compute the join over these $s$ tables. With a slight abuse of notations, we still use "$P$" to denote the design matrix. We also let $[s] = \{1, 2, \ldots, s\}$ for simplicity.

| $T_1$ | |
|---|---|
| $d_1$ | $d_2$ |
| 1 | 1 |
| 2 | 1 |
| 2 | 2 |
| 3 | 3 |

| $T_2$ | |
|---|---|
| $d_2$ | $d_3$ |
| 1 | 1 |
| 1 | 4 |
| 3 | 1 |
| 3 | 3 |

| $T_1 \bowtie T_2$ | | |
|---|---|---|
| $d_1$ | $d_2$ | $d_3$ |
| 1 | 1 | 1 |
| 1 | 1 | 4 |
| 2 | 1 | 1 |
| 2 | 1 | 4 |
| 3 | 3 | 1 |
| 3 | 3 | 3 |

Table 1: An illustration for the join over two tables.

**Definition 1 (Join)** *The join over the given $s$ tables returns a $n \times d$ design matrix $P = T_1 \bowtie \cdots \bowtie T_s$, such that for any vector (point) $p \in \mathbb{R}^d$, $p \in P$ if and only if $\forall i \in [s]$, $\mathrm{Proj}_{D_i}(p) \in T_i$, where $\mathrm{Proj}_{D_i}(p)$ is the projection of $p$ on the subspace spanned by the features of $D_i$.*

Table 1 is a simple illustration for joining two relational tables. To have a more intuitive understanding of join, we can generate a **hypergraph** $\mathcal{G} = (\mathcal{V}, \mathcal{E})$. Each vertex of $\mathcal{V}$ corresponds to an individual feature of $D$; each hyperedge of $\mathcal{E}$ corresponds to an individual relational table $T_i$, and it connects all the vertices (features) of $D_i$. Then we can define the **acyclic** and **cyclic** join queries. A join is acyclic if the hypergraph $\mathcal{G} = (\mathcal{V}, \mathcal{E})$ can be ablated to be empty by performing the following operation iteratively: if there exists a vertex that connects with only one hyperedge, remove this vertex together with the hyperedge. Otherwise, the join is cyclic. A cyclic query usually is extremely difficult and has a much higher complexity than that of an acyclic query. For example, for a cyclic join, it is even NP-hard to determine that whether the join is empty or not [41]. Fortunately, most real-world joins are acyclic, which allow us to take full advantage of relational data. Similar with most of the previous articles on relational learning [3; 4], we also assume that the join of the given tables is acyclic in this paper.

**Counting** is one of the most common aggregation queries on relational data. It returns the number of tuples that satisfy some particular conditions. The counting on an acyclic join (without additive inequalities) can be implemented effectively via dynamic programming [25]. But when the additive inequalities are included, the counting problem can be much more challenging and we may even need to materialize the whole design matrix [3]. For example, to count the tuples of $\{t \in T_1 \bowtie T_2 \ \& \ \|t\|_2^2 \le 5\}$ in Table 1, we need to check the whole design matrix $T_1 \bowtie T_2$ for selecting the tuples that satisfy the constraint "$\|t\|_2^2 \le 5$".

For the machine learning problems studied in this paper, we have the following assumption for the loss function in (1).

**Assumption 1 (Continuity)** *There exist real constants $\alpha, z \ge 0, \beta \in [0, 1)$, such that for any $p, q \in \mathbb{R}^d$ and any $\theta$ in the hypothesis space, we have*

$$|f(\theta, p) - f(\theta, q)| \le \alpha\|p - q\|^z + \beta|f(\theta, q)|, \tag{2}$$

*where $\|\cdot\|$ is the Euclidean norm in the space.*

**Remark 1** *Different machine learning problems have different values for $\alpha, \beta$ and $z$. For example, for the $k$-means clustering, we have $z = 2, \beta = \epsilon$ and $\alpha = O(\frac{1}{\epsilon})$ (where $\epsilon$ can be any small number in $(0, 1)$); for the $k$-median clustering, we have $z = 1, \beta = 0$ and $\alpha = 1$. Actually for a large number of problems, $\beta$ is small or even 0, e.g., the logistic regression and SVM with soft margin problems have the value $\beta = 0$.*

We define the following coreset with both multiplicative error and additive error. We are aware that the standard coresets usually only have multiplicative errors [43; 27]. However, the deviation bounds for the ERM problems with finite training data set only yield additive error guarantees and the additive error is usually acceptable in practice [8; 55]. So the coresets with additive error have been also proposed recently [9]. Let $P = \{p_1, p_2, \ldots, p_n\}$ be the training set (which is not explicitly given), and denote by $\Delta$ the diameter of $P$ (i.e., the largest pairwise distance of $P$).

**Definition 2 ($(\epsilon_1, \epsilon_2)_z$-Coreset)** *Suppose $\epsilon_1 \in (0, 1)$ and $\epsilon_2, z > 0$. The $(\epsilon_1, \epsilon_2)_z$-coreset, denoted as $\tilde{P}$, is a point set $\{c_1, \cdots, c_{|\tilde{P}|}\}$ with a weight vector $W = [w_1, w_2, \ldots, w_{|\tilde{P}|}]$ satisfying that*

$$\tilde{F}(\theta) \in (1 \pm \epsilon_1)F(\theta) \pm \epsilon_2\Delta^z, \tag{3}$$

*for any $\theta$ in the hypothesis space $\mathbb{H}$, where $\tilde{F}(\theta) = \frac{1}{\sum_{i=1}^{|\tilde{P}|} w_i} \sum_{i=1}^{|\tilde{P}|} w_i f(\theta, c_i)$.*

Usually we want the coreset size $|\tilde{P}|$ to be much smaller than $|P|$. So we can run our algorithm on $\tilde{P}$ and save a large amount of running time.

As mentioned in Section 1.1, we also assume that the training set $P$ has a low intrinsic dimension. "Doubling dimension" is a widely used measure for indicating the intrinsic dimension of a given data set [40]. Roughly speaking, it measures the growth rate of the given data in the space. Recently it has gained a lot of attention for studying its relation to coresets and machine learning problems [38; 32; 19]. For any $c \in \mathbb{R}^d$ and $r \geq 0$, we use $\mathbb{B}(c, r)$ to denote the ball centered at $c$ with radius $r$.

**Definition 3 (Doubling Dimension)** *The doubling dimension of a data set $P$ is the smallest number $\rho > 0$, such that for any $p \in P$ and $r \geq 0$, $P \cap \mathbb{B}(p, 2r)$ is always covered by the union of at most $2^\rho$ balls with radius $r$.*

It is easy to obtain the following proposition by recursively applying Definition 3 $\log \frac{\Delta}{r}$ times.

**Claim 1** *For a given data set $P$ and radius $r > 0$, if $P$ has the doubling dimension $\rho$, then it can be covered by $(\frac{\Delta}{r})^\rho$ balls of radius $r$.*

## 3  Relational Coreset via Aggregation Tree

In this section, we present an efficient coreset construction method for relational data. First, we introduce the technique of aggregation tree with pseudo-cube. Actually there is an obstacle for obtaining the coreset from the aggregation tree. The weight of each point of the coreset is determined by the size of its corresponding pseudo-cube; however, the pseudo-cubes may have overlap and it is challenging to separate them in the environment of relational data. To explain our idea more clearly, we temporally ignore this "overlap" issue and show the "ideal" coreset construction result in Section 3.1. Then we show that the overlap issue can be efficiently solved by using random sampling and some novel geometric insights in Section 3.2.

Recall that our input is a set of $s$ relational tables $\{T_1, \ldots, T_s\}$ with the feature (attribute) sets $\{D_1, \ldots, D_s\}$. Also $D = \bigcup_i D_i$ and $|D| = d$. For ease of presentation, we generate a new feature set $\hat{D}_i$ for each $D_i$ as follows: initially, $\hat{D}_1 = D_1$; starting from $i = 2$ to $s$, we let $\hat{D}_i = D_i \backslash (\cup_{l=1}^{i-1} \hat{D}_l)$. It is easy to see that $\bigcup_i \hat{D}_i = D$, and $\forall i, j \in [s], i \neq j, \hat{D}_i \cap \hat{D}_j = \emptyset$. With a slight abuse of notations, we also use $\hat{D}_i$ to represent the subspace spanned by the features of $\hat{D}_i$ (so the Cartesian product $\hat{D}_1 \times \cdots \times \hat{D}_s$ is the the whole space $\mathbb{R}^d$). For any point set $Q \subset \mathbb{R}^d$ (resp., any point $c \in \mathbb{R}^d$) and any subspace $H$ of $\mathbb{R}^d$, we use $\mathrm{Proj}_H(Q)$ (resp., $\mathrm{Proj}_H(c)$) to denote the projection of $Q$ (resp., $c$) onto $H$.

### 3.1  Coreset Construction

Since our coreset construction algorithm is closely related to the Gonzalez's algorithm for $k$-center clustering [29], we briefly introduce it first. Initially, it arbitrarily selects a point from $P$, say $c_1$, and sets $C = \{c_1\}$; then it iteratively selects a new point that is farthest to the set $C$, i.e., $\arg \max_{p \in P} \min_{q \in C} ||p - q||$, and adds it to $C$. After $k$ iterations, we obtain $k$ points in $C$ (suppose $C = \{c_1, \cdots, c_k\}$). The Gonzalez's algorithm yields a 2-approximation for $k$-center clustering. If the optimal radius of $k$-center clustering on $P$ is $r_{\mathsf{opt}}$, then $P$ can be covered by the $k$ balls $\mathbb{B}(c_1, 2r_{\mathsf{opt}}), \cdots, \mathbb{B}(c_k, 2r_{\mathsf{opt}})$. Together with Claim 1, for any given radius $r > 0$, we know that if we run the Gonzalez's algorithm on $P$ with setting $k = (\frac{2\Delta}{r})^\rho$, the set $P$ can be covered by the balls $\mathbb{B}(c_1, r), \cdots, \mathbb{B}(c_k, r)$ (since $r_{opt} \leq r/2$). If we set $r$ to be small enough, the obtained $C$ can be a good approximation (informally a coreset) for $P$.

However, as mentioned in Section 1, such $k$-center clustering approach cannot be implemented on relational data since it is equivalent to an instance of FAQ-AI($k - 1$). A straightfoward idea to remedy this issue is to run the Gonzalez's algorithm in each subspace $\hat{D}_i$, and then compute the Cartesian product to build a grid of size $k^s$ as the methods of [22; 24]. But this approach will result in an exponentially large coreset (e.g., there are $s = 8$ tables and $k = 1000$). Below, we introduce our algorithm that can achieve a coreset with quasi-polynomial size.

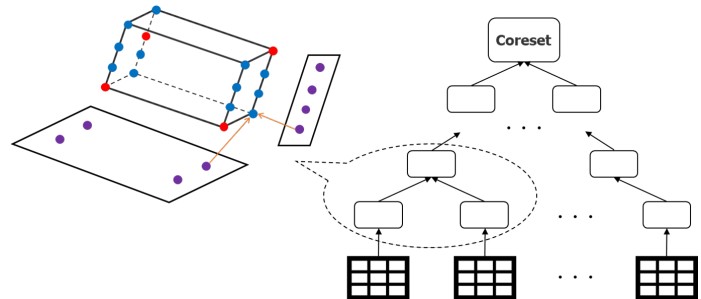

Figure 1: For the example in the figure, $k = 4$. When merging two children, we have $C_{\nu_l}$ and $C_{\nu_r}$ from two disjoint subspaces (the purple points). Then we build the $4 \times 4$ grid, and select $C_{\nu_p}$ (the red points) by running the Gonzalez's algorithm on the grid points.

**Our high-level idea.** We construct a **bottom-to-top tree** $\mathcal{T}$ where each relational table $T_i$ corresponds to a leaf. For each node $\nu \in \mathcal{T}$, it is associated with a set of components: an index set $I_\nu \subseteq [s]$, the spanned subspace $H_\nu = \Pi_{i \in I_\nu} \hat{D}_i$, and a collection of centers $C_\nu = \{c_{\nu,1}, \cdots, c_{\nu,k}\} \subset H_\nu$. The center set $C_\nu$ determines a set of pseudo-cubes (as Definition 4 below) in $H_\nu$. Obviously, for a leaf node (table) $\nu$ of $T_i$, the index set $I_\nu = \{i\}$ and the space $H_\nu = \hat{D}_i$; we directly run the Gonzalez's algorithm on the projection of $P$ on $\hat{D}_i$ to obtain the set $C_\nu$. Then, we grow the tree $\mathcal{T}$ from these leaves. For each parent node $\nu_p$, we compute a grid $C_{\nu_l} \times C_{\nu_r}$ of size $k^2$ by taking the Cartesian product of the center sets from its two children $\nu_l$ and $\nu_r$; then we remove the "empty" grid points (we will explain the meaning of "empty" after Definition 4) and run the Gonzalez's algorithm on the remaining grid points, which are denoted as $\widetilde{C_{\nu_l} \times C_{\nu_r}}$, to achieve the set $C_{\nu_p}$. The index set $I_{\nu_p} = I_{\nu_l} \cup I_{\nu_r}$, and the subspace $H_{\nu_p} = H_{\nu_l} \times H_{\nu_r}$. Finally, the set $C_{\nu_0}$ of the root node $\nu_0$ yields a coreset for the point set $P$. See an illustration of the aggregation tree in Figure 1.

It is worth noting that our aggregation tree approach is fundamentally different to the hierarchical decomposition tree methods that were commonly studied in doubling metrics [30; 12] and other tree structures like $kd$-tree [23]. They build their trees from top to bottom, while our approach is from bottom to top. Actually, our approach can be viewed as a "compressed" version of the grid coreset methods [22; 24], where we deal with a grid of size at most $k^2$ each time. Also, the accumulated error can be also bounded since the height of $\mathcal{T}$ is $O(\log s)$.

**Definition 4 (Pseudo-cube)** *Given an index set $I \subseteq [s]$, a point $c \in H = \prod_{i \in I} \hat{D}_i$, and a number $r \geq 0$, we define the pseudo-cube*

$$\text{PC}_I(c, r) = \prod_{i \in I} \mathbb{B}(\text{Proj}_{\hat{D}_i}(c), r), \tag{4}$$

*where $\text{Proj}_{\hat{D}_i}(c)$ is the projection of $c$ onto the subspace $\hat{D}_i$.*

A nice property of pseudo-cube is that it is a Cartesian product of a set of regions from those subspaces $\hat{D}_i$s. More importantly, it does not involve any cross-table constraints or additive inequalities. Consequently, counting the size of a pseudo-cube is easy to implement over the relational data. Another property is that the Cartesian product of two pseudo-cubes is still a pseudo-cube, but in a new subspace with higher dimension. When we take the grid of the two center sets from two children nodes, we also obtain $k^2$ pseudo-cubes. For each pseudo-cube, we count its size (i.e., the size of its intersection with $P$); if the size is $0$, we call the corresponding grid point is "empty" and remove it before running the Gonzalez's algorithm on the grid.

Let $\text{d}(A, B) := \max_{a \in A} \min_{b \in B} ||a - b||$ for any two sets $A$ and $B$ in $\mathbb{R}^d$ (i.e., the directed Hausdorff Distance). First, for each $h = 0, 1, \cdots, \lceil \log s \rceil$, we define a key value

$$L_h : = \max\{\text{d}(\text{Proj}_{H_\nu}(P), C_\nu) \mid \nu \text{ is at the } h\text{-th level of } \mathcal{T}\}. \tag{5}$$

By using these $L_h$s, we present our coreset construction approach in Algorithm 1.

**Theorem 1** *Suppose the loss function $f(\cdot, \cdot)$ satisfies Assumption 1. If we set $k = \left( \left(\frac{\alpha}{\epsilon_2}\right)^{\frac{1}{z}} 3^{\lceil \log s \rceil} \cdot 2^{\frac{\lceil \log s \rceil^2 + 3\lceil \log s \rceil + 8}{4}} \right)^\rho = \left( \left(\frac{\alpha}{\epsilon_2}\right)^{\frac{1}{z}} 2^{O((\log s)^2)} \right)^\rho$ in Algorithm 1, the returned set $C_{\nu_0}$*

---

**Algorithm 1** AGGREGATION TREE

---

**Input:** A set of relational tables $\{T_1, \ldots, T_s\}$ and a parameter $k$ (the coreset size).
**Output:** A weighted point set as the coreset.

1. Initialize an empty aggregation tree $\mathcal{T}$; when a node $\nu$ of $\mathcal{T}$ is constructed, it is associated with an index set $I_\nu$ and its corresponding subspace $H_\nu$, and a set of $k$ centers $C_\nu = \{c_{\nu,1}, \ldots, c_{\nu,k}\} \subset H_\nu$.

2. Construct $s$ leaf nodes $\{\nu_1, \ldots, \nu_s\}$ corresponding to the $s$ tables at the 0-th level: run the Gonzalez's algorithm to select $k$ centers for each table $T_i$ on the subspace $\hat{D}_i$. Also obtain the value $L_0$ defined in (5).

3. Let $\texttt{Children} = \{\nu_1, \ldots, \nu_s\}$ and $\texttt{Parent} = \emptyset$. Initialize $h = 0$ to indicate the current level on $\mathcal{T}$.

4. For $h = 1$ to $\lceil \log s \rceil$:

  (a) Initialize $l_h = 0$;
  (b) While $|\texttt{Children}| >= 2$:

   i. Select two different nodes $\nu_i, \nu_j$ from $\texttt{Children}$, and $\texttt{Children} = \texttt{Children} \setminus \{\nu_i, \nu_j\}$;
   ii. Compute the Cartesian product to construct a grid $C_{\nu_i} \times C_{\nu_j}$ with $k^2$ centers in the space $H_{\nu_i} \times H_{\nu_j}$;

   iii. For each $c \in C_{\nu_i} \times C_{\nu_j}$, perform the operation "COUNT" to count the size of $\Big\{ p \in P \mid$ $\text{Proj}_{H_{\nu_i}}(p) \in \texttt{PC}_{I_{\nu_i}}(\text{Proj}_{H_{\nu_i}}(c), L_{h-1}) \ \& \ \text{Proj}_{H_{\nu_j}}(p) \in \texttt{PC}_{I_{\nu_j}}(\text{Proj}_{H_{\nu_j}}(c), L_{h-1}) \Big\}$;

   iv. Remove $c$ from $C_{\nu_i} \times C_{\nu_j}$ if the size is 0, and then obtain $\widetilde{C_{\nu_i} \times C_{\nu_j}}$ that is the set of remaining non-empty grid points;

   v. Construct a parent node $\nu$: run the Gonzalez's algorithm on $\widetilde{C_{\nu_i} \times C_{\nu_j}}$ to obtain a set $C_v$ of $k$ centers, let $l_h = \max(l_h, \texttt{d}(\widetilde{C_{\nu_i} \times C_{\nu_j}}, C_\nu))$, $I_\nu = I_{\nu_i} \cup I_{\nu_j}$ and $H_\nu = H_{\nu_i} \times H_{\nu_j}$;
   vi. $\texttt{Parent} = \texttt{Parent} \cup \{\nu\}$;

  (c) $L_h = \sqrt{2^h}(l_h + \sqrt{2}L_{h-1})$;
  (d) $\texttt{Children} = \texttt{Children} \cup \texttt{Parent}, \texttt{Parent} = \emptyset$;

5. Let $\nu_0$ be the root node; for each $c_{\nu_0, i} \in C_{\nu_0}$, we assign a weight $w_i = |P \cap \texttt{PC}_{I_{\nu_0}}(c_{\nu_0, i}, L_{\lceil \log s \rceil})|$ (actually, this is not an accurate expression since the pseudo-cubes may have overlap with each other, and we will address this issue in Section 3.2).

  **return** the set $C_{\nu_0}$ with the weights $\{w_1, \cdots, w_k\}$.

---

with the weights $\{w_1, \cdots, w_k\}$ yields a $(\beta, \epsilon_2)_z$-coreset. The time complexity of constructing the tree $\mathcal{T}$ is $O\left(sk^2\Psi(N, s, d) + sk^3\right)$, where $\Psi(N, s, d)$ is the complexity of performing the counting each time [1] (for counting an acyclic join, $\Psi(N, s, d) = O\left(sd^2 N \log(N)\right)$).

**Remark 2** *Note that our coreset size is quasi-polynomial since it has a factor $2^{O((\log s)^2)}$. But we would like to emphasize that this factor usually is small for most practical problems. For example, in the standard TPC-H benchmark [47], $s$ is no larger than $8$, although the design matrix over joining the $s$ tables can be quite large.*

To prove the above theorem, we first introduce the following key lemma. We let $r_0 = \frac{\Delta}{k^{\frac{1}{\rho}}}$. From Claim 1 we know that the entire data set $P$ can be covered by $k$ balls with radius $r_0$.

**Lemma 1** *For each $h = 0, 1, \cdots, \lceil \log s \rceil$, $L_h \le 3^h \cdot 2^{\frac{h^2 + 3h + 8}{4}} r_0$.*

*Proof.* Since the Gonzalez's algorithm yields an approximation factor 2, from Claim 1 we know that $L_0 \le 2r_0$. Then we consider the case $h \ge 1$. Suppose the algorithm is constructing the $h$-th level of $\mathcal{T}$. In Step 4b we repeatedly select two nodes $\nu_i, \nu_j$ from the $(h-1)$-th level that form a parent node $\nu$ at the $h$-th level. Let $C_{opt}$ be the optimal solution of the $k$-center clustering on $\text{Proj}_{H_{\nu_i} \times H_{\nu_j}}(P)$. Since $P$ can be covered by $k$ balls with radius $r_0$, so can the projection $\text{Proj}_{H_{\nu_i} \times H_{\nu_j}}(P)$. That is,

$$\texttt{d}(\text{Proj}_{H_{\nu_i} \times H_{\nu_j}}(P), C_{opt}) \le r_0. \tag{6}$$

Also, since $\mathtt{d}(\mathrm{Proj}_{H_{\nu_i}}(P), C_{\nu_i}) \leq L_{h-1}$ and $\mathtt{d}(\mathrm{Proj}_{H_{\nu_j}}(P), C_{\nu_j}) \leq L_{h-1}$, we have

$$\mathtt{d}(\mathrm{Proj}_{H_{\nu_i} \times H_{\nu_j}}(P), \widetilde{C_{\nu_i} \times C_{\nu_j}}) \leq \sqrt{2}L_{h-1}. \tag{7}$$

Note that we use $\widetilde{C_{\nu_i} \times C_{\nu_j}}$ instead of $C_{\nu_i} \times C_{\nu_j}$ in the above (7). The reason is that the Cartesian product $C_{\nu_i} \times C_{\nu_j}$ may contain some "empty" grid points (see Algorithm 1 step 4(b)(iii)), and the distance bound "$\sqrt{2}L_{h-1}$" does not hold for them.

Together with (6), we know

$$\mathtt{d}(\widetilde{C_{\nu_i} \times C_{\nu_j}}, C_{opt}) \leq r_0 + \sqrt{2}L_{h-1}. \tag{8}$$

That is, if we run $k$-center clustering on $\widetilde{C_{\nu_i} \times C_{\nu_j}}$, the optimal radius should be no larger than $r_0 + \sqrt{2}L_{h-1}$. So if we run the 2-approximation Gonzalez's algorithm on $\widetilde{C_{\nu_i} \times C_{\nu_j}}$ and obtain the center set $C_\nu$, then

$$\mathtt{d}(\widetilde{C_{\nu_i} \times C_{\nu_j}}, C_\nu) \leq 2(r_0 + \sqrt{2}L_{h-1}). \tag{9}$$

Further, we combine (7) and (9), and obtain

$$\mathtt{d}(\mathrm{Proj}_{H_{\nu_i} \times H_{\nu_j}}(P), C_\nu) \leq 2(r_0 + \sqrt{2}L_{h-1}) + \sqrt{2}L_{h-1}. \tag{10}$$

For each $c_{\nu,l} \in C_\nu$, we construct a pseudo-cube

$$\prod_{t \in I_{\nu_i} \cup I_{\nu_j}} \mathbb{B}(\mathrm{Proj}_{\hat{D}_t}(c_{\nu,l}), 2(r_0 + \sqrt{2}L_{h-1}) + \sqrt{2}L_{h-1}). \tag{11}$$

Then we know that $\mathrm{Proj}_{H_{\nu_i} \times H_{\nu_j}}(P)$ is covered by the union of the obtained $k$ pseudo-cubes. Because $|I_\nu| = |I_{\nu_i} \cup I_{\nu_j}| \leq 2^h$, we have $L_h \leq (2(r_0 + \sqrt{2}L_{h-1}) + \sqrt{2}L_{h-1}) \cdot \sqrt{2^h}$. Together with $L_0 \leq 2r_0$, we can solve this recursion function and obtain $L_h \leq 3^h \cdot 2^{\frac{h^2 + 3h + 8}{4}} r_0$. $\qquad\square$

Lemma 1 indicates that the difference between $C_{\nu_0}$ and $P$ is bounded in the space. To prove it is a qualified coreset with respect to the ERM problem (1), we also need to assign a weight to each point of $C_{\nu_0}$. In Step 5 of Algorithm 1, we set the weight $w_i = |P \cap \mathtt{PC}_{I_{\nu_0}}(c_{\nu_0,i}, L_{\lceil \log s \rceil})|$ (for simplicity, we temporarily assume these pseudo-cubes are disjoint, and leave the discussion on the overlap issue to Section 3.2). The detailed proof of Theorem 1 is placed to our full paper [14].

**Remark 3** *In a real implementation, if the coreset size is given (e.g., $5\%$ of the input data size $|P|$), we can directly set the value for $k$ to run Algorithm 1. Another scenario is that we are given a restricted variance between $P$ and the output $C_{\nu_0}$ (i.e., the difference $\mathtt{d}(P, C_{\nu_0})$ is required to be no larger than a threshold). Then we can try the value for $k$ via doubling search. For example, starting from a small constant $k_0$, we can try $k = k_0, 2k_0, 2^2 k_0, \cdots$, until the difference is lower than the threshold.*

## 3.2 The Overlap Issue

We consider the $k$ pseudo-cubes $\mathtt{PC}_{I_{\nu_0}}(c_{\nu_0,i}, L_{\lceil \log s \rceil})$, $1 \leq i \leq k$, obtained from Algorithm 1. Since $\nu_0$ is the root, it is easy to know $I_{\nu_0} = [s]$ and $H_{\nu_0} = \mathbb{R}^d$. For the sake of simplicity, we use $\mathtt{PC}_i$ to denote the pseudo-cube $\mathtt{PC}_{I_{\nu_0}}(c_{\nu_0,i}, L_{\lceil \log s \rceil})$ and $L$ to denote $L_{\lceil \log s \rceil}$ below.

If these pseudo-cubes are disjoint, we can simply set the weight $w_i = |P \cap \mathtt{PC}_i|$ as Step 5 of Algorithm 1. However, these pseudo-cubes may have overlaps and such an assignment for the weights cannot guarantee the correctness of our coreset. For example, if a point $q \in P$ is covered by two pseudo-cubes $\mathtt{PC}_{i_1}$ and $\mathtt{PC}_{i_2}$, we should assign $q$ to only one pseudo-cube. Note that this overlap issue is trivial if the whole data matrix $P$ is available. But it can be troublesome for relational data, because it is quite inefficient to perform the counting on their union $P \cap (\mathtt{PC}_{i_1} \cup \mathtt{PC}_{i_2})$. Moreover, the $k$ pseudo-cubes can partition the space into as large as $2^k$ different regions, and it is challenging to deal with so many overlaps.

Fortunately, we can solve this issue by using random sampling. The key observation is that we can tolerate a small error on each weight $w_i$. Let $\delta \in (0, 1)$. If we obtain a set of approximate weights $W' = \{w_1', \cdots, w_k'\}$ that satisfy $w_i' \in (1 \pm \delta)w_i$ for $1 \leq i \leq k$, then we have

$$\sum_{i=1}^{k} w_i' f(\theta, c_{\nu_0,i}) \in (1 \pm \delta) \sum_{i=1}^{k} w_i f(\theta, c_{\nu_0,i}), \tag{12}$$

for any $\theta$ in the hypothesis space $\mathbb{H}$. We can use the following idea to obtain a qualified $W'$.

**High-level idea.** Without loss of generality, we assume all the $k$ pseudo-cubes are not empty (otherwise, we can directly remove the empty pseudo-cubes). Then we consider the pseudo-cubes one by one. For $\mathtt{PC}_1$, we directly set $w_1' = |P \cap \mathtt{PC}_1|$. Suppose currently we have already obtained the values $w_1', \cdots, w_{i_0}'$, and try to determine the value for $w_{i_0+1}'$. We take a uniform sample of $m$ points from $P \cap \mathtt{PC}_{i_0+1}$ by using the sampling technique for relational data [59]. Each sampled point corresponds a binary random variable $x$: if it belongs to $\mathtt{PC}_{i_0+1} \setminus (\cup_{i=1}^{i_0} \mathtt{PC}_i)$, $x = 1$; otherwise, $x = 0$. If $m$ is sufficiently large, from the Chernoff bound, we can prove that the sum of these $m$ random variables (denote by $g$) over $m$ can serve as a good estimation of

$$\tau_{i_0+1} = \frac{|P \cap (\mathtt{PC}_{i_0+1} \setminus (\cup_{i=1}^{i_0} \mathtt{PC}_i))|}{|P \cap \mathtt{PC}_{i_0+1}|}. \tag{13}$$

Thus we can set $w_{i_0+1}' = \frac{g}{m} \cdot |P \cap \mathtt{PC}_{i_0+1}| \approx |P \cap (\mathtt{PC}_{i_0+1} \setminus (\cup_{i=1}^{i_0} \mathtt{PC}_i))|$.

A remaining issue of the above method is that the ratio $\tau_{i_0+1}$ can be extremely small, that is, we have to set $m$ to be too large for guaranteeing the multiplicative "$1 \pm \delta$" error bound. Our idea for solving this issue is from the geometry. If $\tau_{i_0+1}$ is extremely small, we label $\mathtt{PC}_{i_0+1}$ as a "light" pseudo-cube. Meanwhile, it should has at least one "heavy" neighbor from $\{\mathtt{PC}_1, \cdots, \mathtt{PC}_{i_0}\}$. The reason is that a small $\tau_{i_0+1}$ implies that there must exist some $1 \leq i_1 \leq i_0$, such that the intersection $P \cap \mathtt{PC}_{i_0+1} \cap \mathtt{PC}_{i_1}$ takes a significant part of $P \cap \mathtt{PC}_{i_0+1}$. And we label $\mathtt{PC}_{i_1}$ as a "heavy" neighbor of $\mathtt{PC}_{i_0+1}$. Moreover, the distance between $\mathtt{PC}_{i_1}$ and $\mathtt{PC}_{i_0+1}$ should be bounded (since they have overlap in the space). Therefore, we can just ignore $\mathtt{PC}_{i_0+1}$ and use $\mathtt{PC}_{i_1}$ to represent their union $\mathtt{PC}_{i_1} \cup \mathtt{PC}_{i_0+1}$. This idea can help us to avoid taking a too large sample for light pseudo-cube. Overall, we have the following theorem, and the detailed proof is placed to our full paper [14].

**Theorem 2** *Suppose the loss function $f(\cdot, \cdot)$ satisfies Assumption 1 and $\epsilon_1 > \beta$. If we set $k$ in Algorithm 1 as Theorem 1, and let $m \geq \Theta(\frac{k^2}{(\epsilon_1 - \beta)^2 \epsilon_1} \log \frac{k}{\lambda})$ where $\lambda \in (0, 1)$, the returned set $C_{\nu_0}$ with the weights $W'$ yields a $(\epsilon_1, \epsilon_2)_z$-coreset with probability $1 - \lambda$. The sampling procedure takes $O(k^2 md)$ time.*

## 4  Experimental Evaluation

We evaluate the performance of our relational coreset on three popular machine learning problems, the SVM with soft margin ($\alpha = O(\|\theta\|_2)$, $\beta = 0$, and $z = 1$), the $k_c$-means clustering[3] ($\alpha = O(\frac{1}{\epsilon})$, $\beta = \epsilon$, and $z = 2$, where $\epsilon$ can be any small number in $(0, 1)$), and the logistic regression ($\alpha = O(\|\theta\|_2)$, $\beta = 0$, and $z = 1$). All the experimental results were obtained on a server equipped with 3.0GHz Intel CPUs and 384GB main memory. Our algorithms were implemented in Python with PostgreSQL 12.10. We release our codes at Github [28].

**Data sets and Queries.** We design four different join queries (Q1-Q4) on three real relational data sets. Q1 and Q2 are designed on a labeled data set HOME CREDIT [31], and we use them to solve the SVM and logistic regression problems. Q3 and Q4 are foreign key joins [5] designed on the unlabeled data sets YELP [58] and FAVORITA [26] respectively, and we use them to solve the $k_c$-means clustering problem.

**Baseline methods.** We consider five baseline methods for comparison. (1) ORIGINAL: construct the complete design matrix $P$ and run the training algorithm directly on it. (2) ORI-GON: construct $P$ as ORIGINAL, run the Gonzalez's algorithm [29] on $P$ to obtain the centers, and then run the training algorithm on the centers. (3) UNIFORM: the relational uniform sampling algorithm [59]. (4) R$k$-MEANS: the relational $k_c$-means algorithm [22]. (5) RCORE: our relational coreset approach.

---

[3] We use "$k_c$" instead of "$k$" to avoid being confused with the coreset size $k$

We consider both the running time and optimization quality. We record the end-to-end runtime that includes the design matrix/coreset construction time and the training time. For the optimization quality, we take the objective value $F(\theta^*)$ obtained by ORIGINAL as the optimal objective value; for each baseline method, we define "**Approx.**"$= \frac{F(\theta)-F(\theta^*)}{F(\theta^*)}$, where $F(\theta)$ is the objective value obtained by the method.

| Coreset size | | 200 | 400 | 600 | 800 | 1000 |
|---|---|---|---|---|---|---|
| **End-to-end runtime (s)** | ORIGINAL | | | $> 21600$ | | |
| | ORI-GON | 3808 | 5208 | 6606 | 8044 | 9434 |
| | UNIFORM | 34 | 35 | 35 | 36 | 38 |
| | RCORE | 208 | 288 | 363 | 446 | 531 |
| **Approx.** | ORI-GON | 1.41 | 1.50 | 1.29 | 1.12 | 0.95 |
| | UNIFORM | 2.22 | 2.60 | 2.09 | 2.23 | 2.14 |
| | RCORE | 0.92 | 0.31 | 0.27 | 0.16 | 0.02 |

Table 2: The results of SVM on Q1.

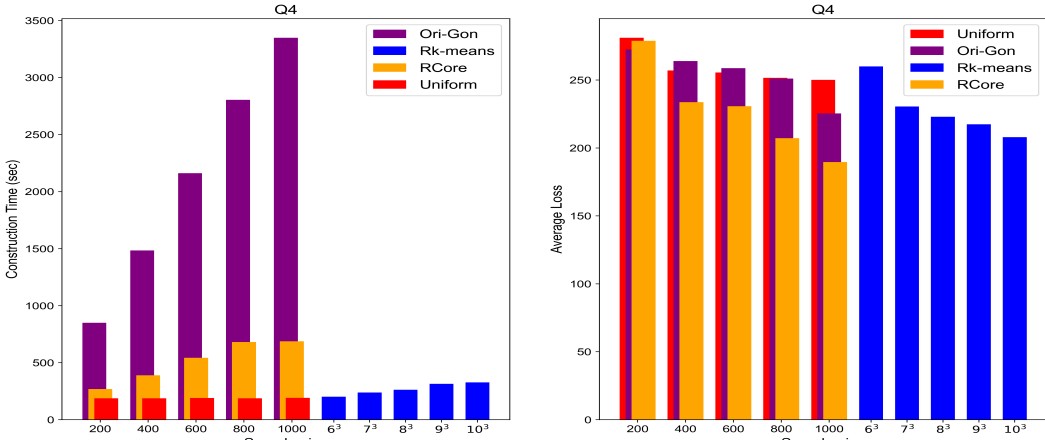

Figure 2: The results of $k_c$-means on Q4.

Due to the space limit, we only present part of the results (Table 2 and Figure 2) here. In general, ORIGINAL and ORI-GON are the most time-consuming ones because they need to construct the whole design matrix and perform the operations (like train the models or run the Gonzalez's algorithm) on the data (even if the queries are foreign key joins). UNIFORM is always the fastest one, because it only takes the simple uniform sampling procedure; but its overall optimization performance is relatively poor. Except ORIGINAL, our RCORE has the best optimization quality for most cases with acceptable running time. We leave the more detailed experimental results to our full paper [14].

## 5 Conclusion

In this paper, we propose a novel relational coreset method based on the aggregation tree with pseudo-cube technique. Our method can be applied to several popular machine learning models and has provable quality guarantees. In future, there are also several interesting problems deserved to study, e.g., the relational coresets construction for more complicated machine learning models, and the privacy-preserving issues for relational coresets when the input data contains sensitive attributes.

## 6 Acknowledgements

The research of this work was supported in part by National Key R&D program of China through grant 2021YFA1000900 and the Provincial NSF of Anhui through grant 2208085MF163. We also want to thank the anonymous reviewers for their helpful comments.

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
