# OpenReview forum: "Coresets for Relational Data and The Applications"
_NeurIPS.cc/2022/Conference — NeurIPS 2022 Accept_

### Official Review · Reviewer_ZRDo · 2022-07-07

**Rating:** 7
**Confidence:** 2
**Soundness:** 3 good
**Presentation:** 3 good
**Contribution:** 4 excellent

**Summary:**

The paper proposes an algorithm for constructing a coreset from relational data without materializing the data matrix by joining the tables. The paper improves over previous work by considering ML tasks other than clustering and by avoiding a coreset size exponential in the number of tables. The produced coresets have a bound on the error on the empirical risk that has both a multiplicative and an additive component.

The result is a achieved by using an aggregation tree that builds the coreset bottom up from the individual tables.
At each level of the tree Gonzalez's clustering algorithm for $k$-center clustering is applied to obtain local coresets that are then combined in the parent node building so called pseudo-cubes. The weights of the points in the final coreset
are obtained by counting the intersection of pseudo-cubes with the original dataset. To avoid overcounting due to pseudo-cubes overlapping, sampling is used.

The supplementary material shows experiments on three datasets for k-means clustering, SVM and logistic regression.
The speedup obtained for SVM and LR with respect to the materialization case is of one or two order of magnitude.
In clustering, the comparison with R$k$-means show that the approach is advantageous when there are more than 3 tables both in terms of time and of solution quality.

**Questions:**


1) Please define the concepts mentioned above
2) Please provide a proof of Claim 1
3) Please explain how the fact that nodes from different level may be combined does not pose problems for the proofs.
4) In the experiments, the Home Credit dataset has binary attributes, not real attributes. Please explain why this does not pose problems with your method.

**Limitations:**

Yes, the limitations of the approach has been presented. The work does not have negative social impacts.

**Strengths And Weaknesses:**

The difference with related work is clearly highlighted, in particular with the R$k$-means algorithm of [Curtin et al 2020]: the proposed algorithm tackles other tasks besides clustering, is not limited to coresets exponential in the size of the number of tables and the experiments show it is faster, so the originality is high.

Technically the paper appears sound, even if it is difficult to check for a non expert in the domain because of the use of specific terminology that is not introduced:
1) In equation 3, $\Delta$ appears without definition
2) From line 185 we learn that $\Delta$ is the diameter of the dataset, but the concept of diameter is not defined.
3) Claim 1 is non obvious and should be proved. Moreover, the number $(\frac{\Delta}{r})^\rho$ may be non integer, how do you deal with such a case?
4) Line 211 speaks of the optimal radius of $k$-center clustering without defining it
Another technical problem is the assumption that a node at level h of the aggregation tree is built from two nodes at level h-1: however, if the number of tables is not a power of 2, at one point you will have to combine two nodes at different levels. Does this poses a problem to the proofs?

The presentation is overall clear, apart from the undefined concept mentioned above. "Additive inequalities" should be defined as well. In the algorithm, the variable $Parent$ is unnecessary: the algorithm would be simpler and easier to understand if point (d) would simply be $Children=Children \cup \{ \nu \}$.

The significance of the contribution is high, as the speedups that can be obtained are impressive especially for SVM and LR.

---

> ### Author Response · Authors · 2022-08-02
> **Response to Reviewer ZRDo**
>
> We thank the reviewer for the thoughtful comments and suggestions. Our responses and clarifications are summarized below.
>
> **''1. Please define the concepts mentioned above''**
>
> (1) The diameter  $\Delta$ of a dataset $P$ is the distance between the farthest two points in the dataset, i.e., $\Delta = \max\lbrace||p_i - p_j||_2, p_i, p_j \in P\rbrace$.
>
> (2) The optimal radius is the minimum radius for $k$-center clustering, i.e., if we want to cover the point set by $k$ equal-size balls, it is the minimum radius that we need.
>
> (3) ''additive inequalities'': this definition is from [29], where each additive inequality defines a constraint for the query; for example, in SVM, the ''positive'' class is defined by the inequality $\langle x, \omega\rangle>0$, where $\omega$ is the normal vector of the separating hyperplane.
>
>
>
> **''2. Please provide a proof of Claim 1''**
>
> Initially,  we know that the dataset can be covered by the union of at most $2^\rho$ balls with radius $\frac{\Delta}{2}$ based on Definition $3$ (because the dataset can be covered by a ball with radius $\Delta$). Then, we apply this idea to each of these $2^\rho$ balls; that is, the dataset can be covered by the union of at most $2^{2\rho}$ balls with radius $\frac{\Delta}{4}$. If we apply this idea $\log\frac{\Delta}{r}$ times,  we have at most $(\frac{\Delta}{r})^\rho$ balls of radius $r$ in total.
>
>
> **''3. Please explain how the fact that nodes from different level may be combined does not pose problems for the proofs.''**
>
>  Thanks for this question.
> It is not a problem if the number of tables is not a power of 2. For example, if $s=3$, we assume the three sets of $k$ centers from the three tables are $C_{v_1}$, $C_{v_2}$, and $C_{v_3}$, respectively. First, we combine $C_{v_1}$ and $C_{v_2}$, and suppose the obtained new set of $k$ centers is $C_\mu$. Then, we combine $C_\mu$ and $C_{v_3}$. Note that  we only require  that the two combined nodes  represents two orthogonal subspaces, regardless of their levels in the tree.
>
>
>
> **''4. In the experiments, the Home Credit dataset has binary attributes, not real attributes.....''**
>
>  The $\mathrm{Home\ Credit}$ dataset has the binary attribute that indicates  the labels of the data items (the  other binary attributes are just treated as general features). In our experiments, we calculate the corresponding coresets separately for different labels.

---

> > ### Comment · Reviewer_ZRDo · 2022-08-04
> > **comments answered my questions**
> >
> > The authors' comments answered my questions so I will raise my evaluation to Accept

---

### Official Review · Reviewer_FGi9 · 2022-07-11

**Rating:** 7
**Confidence:** 2
**Soundness:** 3 good
**Presentation:** 2 fair
**Contribution:** 3 good

**Summary:**

The paper addresses the problem of finding a coreset for a given
dataset and objective function, i.e., a (smaller and weighted) set of
data points s.t. the objective on those points is always bounded
from below and above by a linear combination of the objective
value of the dataset and a suitable power of the data set radius.
The paper addresses esp. the coreset problem, when the dataset
is relational and given only implicitly as set of yet unjoined
relations, and materializing the join is too expensive. The authors
propose an algorithm that generalizes the well-known initialization
scheme of k-means-clustering, to select the next cluster center as
the data point with maximal distance to all previously selected
ones. Their algorithm starts from coresets of the individual relations
and then consecutively combines two subspaces by computing the
coresets of the cartesian products of the coresets of the two
component spaces. Each core point gets as weight the number
of data points closest to it.

They prove that for objectives whos variation is bound from above
by a linear combination of a power of the L2-distance and the value
of one of the two elements, and a carefully chosen coreset size their
algorithm actually yields a coreset

In the appendix, the authors compare their method against materialization
and rk-means [3] and find their method to be both, faster and more
accurate.


**Questions:**

The problem addressed in this paper looks interesting, but not easy to
tackle. The proposed algorithm follows a plausible overall approach
and its correctness could be proven.

To me, as an outsider of this research direction, the paper and its
contributions look fine.

What is not clear to me: Alg. 1 in principle should also deliver correct
results for the (trivial?) edge case s=1. But in this case it just yields the
data points choosen as initial k-means cluster centers. What good
properties do they have say for logistic regression, as guartanteed
by your theorem 1? Maybe explaining the algorithm from this angle
may make it more accessible to people outside your immediate field?

Small points:
- p. 3 "D=\bigunion_i D_i": I found that initially difficult to understand as
  it is never said what "D_i" is and it later on is used in expressions such as
  "proj_{D_i}" where you usually would expect a space. Saying that "D_i"
  here denotes a set of (usually overlapping) column names would be helpful.
- eq. 3: "\Delta" is only defined on the next page.
  "F(\theta)" is undefined.
- p.5 "optimal radius" is not explained.


**Limitations:**

Yes.

**Strengths And Weaknesses:**

strong points:
s1. the problem addressed looks pretty complicated.
s2. the proposed algorithm follows a plausible overall approach and is
  provided in detail.
s3. the algorithm is proven to be correct; esp. a coreset size is derived.

weak points:
w1. the paper is dense and not easy to follow.
w2. experiments can only be found in the appendix.
w3. the (way simpler?) special case of a single relation is not 100% clear.

---

> ### Author Response · Authors · 2022-08-02
> **Response to Reviewer FGi9**
>
> We thank the reviewer for the thoughtful comments and suggestions. Our responses and clarifications are summarized below.
>
>
> **''What is not clear to me: Alg. 1 in principle should also deliver correct results for the (trivial?) edge case s=1......''**
>
> When $s=1$, our algorithm just returns the $k$-center clustering result on the single table by the Gonzalez's algorithm. Actually, this $k$-center clustering based coreset has been studied before [28, 46, 19] (see our comment in line 92-93). But its realization for  multiple relational tables has never been proposed before, due to several significant challenges studied in the research of '' functional aggregate queries (FAQ)'' (see line 94-98).  Our main contribution is to show that it is possible to tackle these challenges by proposing a novel ''aggregation tree with pseudo-cube'' algorithm.
>
>
> **Minor points:**
>
> 1) For the $D_i$s, we explain their meanings with details in line 197-204 before presenting our algorithm. Thanks for this question, and we will move this part to page 3 (it may be better for reading).
>
> 2) For $F(\theta)$, we define it in the formula (1) (line 65-67).
>
> 3) The optimal radius is the minimum radius for $k$-center clustering, i.e., if we want to cover the point set by $k$ equal-size balls, it is the minimum radius that we need.

---

### Official Review · Reviewer_tvSG · 2022-07-11

**Rating:** 6
**Confidence:** 4
**Soundness:** 3 good
**Presentation:** 3 good
**Contribution:** 3 good

**Summary:**

The manuscript tackles the problem of learning a hypothesis that minimises the the mean loss value of all training data points. The data points are in this case only implicitly given as an acyclic join of relational tables. The manuscript proposes how to select a (possibly weighted) subset of the data points (core set) that is sufficiently representative for approximate solutions of the minimisation problem (multiplicative error $\epsilon_1$, additive error $\epsilon_2$), but aims to achieve significant speed-ups due to the reduced data size. The proposed approach traverses the relational tables directly without materialising the design matrix or join result.

**Questions:**

Q1 What percentage of relational datasets (e.g., in the referenced Kaggle 2017 "State of Data Science Survey") do not allow to materialise the design matrix due to prohibitive computation/storage costs (using mainstream relational database systems on modern hardware)? This question relates to W1 & W2.

Q2 It is argued on l. 39 that the join of tables can be as large as $O(N^s)$. In case of foreign-key joins (see W1) the join of the tables can of course not exceed $N$, but the join result can certainly be vastly larger when joining along non-keys or if it's a cross join (cartesian product). Do join queries with result size in $O(N^s)$ commonly arise in practice (excluding cross joins that likely allow for much simpler methods)? This question relates to W2.

Q3 Suppose it would be feasible to efficiently materialise the join, what are the remaining challenges due to the relational data, e.g., would it be possible to directly apply Gonzalez's algorithm or would there still be an overlap issue? This question relates to W2.

Q4 Which SQL queries are supported over the base tables? This question relates to W3.

Q5 Why are they called "pseudo cubes"? Fig. 1 seems to allude to the geometrical analogy, but it is not completely clear what it aims to explain / how it aims to help. This question relates to W4.

**Ethics Review Area:**

["I don’t know"]

**Limitations:**

The experimental study from the supplemental material is not discussed much in the main paper and seems to be rather limited, i.e., it does not even seem to provide a proof of concept of the proposed methods (see W1). Thus, it would be useful to extend the empirical study to cover queries that would motivate the proposed approach better and add the derived insights in the main paper.

While the work commonly talks about "relational data" which raises the expectation of supporting any type of SQL query it seems to be limited to a narrow set of operators (see W3). It would therefore be good to clarify which subclass of relational queries is supported by the proposed approach and why it is a sufficiently important subclass (see Q4).

**Strengths And Weaknesses:**

**Strengths**

S1 The general problem of the paper seems well-motivated. The premise of the paper is very interesting.

The authors convincingly argue for the importance of relational data in the context of machine learning and core sets are a practical tool to efficiently deal with larger sets of data in a way that formally bounds the utility loss.

S2 There are some potentially significant and non-trivial results that could be also useful beyond the targeted scope

Although it is difficult to fully understand all ideas (see W4), the general ideas seem interesting and may be useful beyond coreset construction.

S3 Apart from the concerns of W1-W5, the general quality of the paper is good (presentation, soundness, non-trivial contributions, relevant topic)

The paper is generally well-written, it touches upon many interesting results in prior work and reasonably uses existing ideas (Gonzalez's algorithm, doubling dimension).

**Weaknesses**

W1 Central claim from the abstract (l.7: "it is expensive to directly materialize the data matrix by joining the tables") is not substantiated

Most relational queries only feature "foreign key joins" which do not pose much of a challenge computation-wise (see Definition 4.1 in "Join Synopses for Approximate Query Answering" by Achyara et al @ SIGMOD'99). For instance, all 3 queries in the experimental study (supplementary material) appear to be "foreign key joins" and the TPC-H benchmark (widely used in database literature) also only features foreign key joins. Note that the authors of [52] had to come up with their own queries ("QX", "QY" etc) that are not part of the TPC-H benchmark to motivate their work over the TPC-H data. Thus, it would be crucial to demonstrate that it is actually too expensive materialize the data matrix for common types of queries, i.e.:

- Consider relational queries with joins along non-key columns with many-to-many-relations (e.g., quasi-identifiers in record linkage).
- Report excessive runtimes (lower bounds suffice) of join materialisation using some modern relational database management system

Some more details on foreign key joins:

Big relational tables are full of redundancies which is extremely inconvenient for database updates and wastes a lot of space. Thus, database normalization splits these big tables into smaller tables. One of these smaller tables ("main table") has the same number of rows as the big table. By joining that table with the other tables along their "foreign key" column values, one reconstructs the big table. Thus, foreign-key joins can be thought of as the inverse operation of database normalization.  For instance in the experimental study (supplementary material) the table "Application" serves as the main table for Q1 & Q2 and "Review" serves as the main table for Q3 ("main table" has same number of rows as join result). Materialising foreign key joins can be done extremely efficiently, especially with main memory databases  (e.g., see the online demo at https://hyper-db.de/interface.html that demonstrates this quite convincingly). It is also trivial to sample from foreign key joins (see Achyara et al @ SIGMOD'99 ; one first samples the main table and then joins the main table sample with the other tables). Join sampling as in [52] is not particularly beneficial for foreign key joins, because it requires prebuilt index structures for all base tables. Thus, if the work is aimed at foreign key joins, it would be useful to adapt the approach to it.

W2 Motivation unclear if join result can actually be materialised (see W1)

If the join can be materialised, there is no "relational data problem" as one can simply compute the design matrix and it is becomes unclear what the manuscript does differently compared to prior work (and it already seems to heavily rely on existing ideas such as Gonzalez’s algorithm).

W3 The proposed approach only supports a narrow subset of relational queries.

The proposed approach inherits the limitations of [52] which for instance does not support arbitrary selections over the join result such as the widely used DISTINCT operator. In contrast, materialising the join allows all operations supported by relational database systems.

W4 The presentation could be improved on some central parts of the paper

For instance the description of the "high-level" idea (l. 222 and following) does not provide a lot of intuition (apart from "bottom to top" vs "top to bottom") and relies a lot on  technical statements. It is often not clear how one sentence relates to the last, i.e., an overarching logical narrative seems to be missing here. Similarly, the naming and description of the "pseudo-cube" is a bit confusing. The description of the algorithm complexity is also difficult to read (l. 343).

W5 The work is not particularly conclusive

Due to a lack of theoretical/empirical baselines it is difficult to judge how useful the proposed techniques are compared to naive approaches (e.g., full materialization) and how much room for improvement is left by the proposed techniques.

W6 Minor issues

- l.117 What are the (not purely theoretical) insights from Chen and Yi [13] with regards to specific cyclic join scenarios and what about relaxing a cyclic join to an acyclic one as in [52] and reject any results that violate the join conditions that complete the cycle?
- l.192 "pesudo-cube" => "pseudo-cube"
- l.318 "we call [...] as a ''light pseudo-cube''" => "we refer to [...] as a ''light pseudo-cube''"

Purely a suggestion: using $\varepsilon$ (varepsilon) instead of $\epsilon$ (epsilon); the supplement references 4 papers, but two of them, i.e., [1] and [2], are not referenced or discussed in the main paper. Perhaps it would be useful to reference and discuss them in the main paper as well.

**Edit after author rebuttal / reviewer discussion**

Overall recommendation raised from reject to weak accept, presuming that the authors fulfil their commitments to address the major concerns in the manuscript as discussed:

* Add baseline *"join-then-Gonzalez"* to experimental evaluation that joins the relations and then runs something akin to Gonzalez's algorithm to obtain a coreset (also add uniform random sampling as a naive baseline)
* Add a query over the missing public dataset from the Rkmeans work [20] and report all results for all queries (which would reveal when exactly the new methods offer a substantial benefit over *"join-then-Gonzalez"*)
* Clarify that extremely common problem instances (foreign key joins) do not pose much of a challenge as the join size cannot exceed the table sizes and could also be handled with a simple combination of existing methods such as *"join-then-Gonzalez"* (thus, the focus of the work as in prior works is on harder problem instances that are also featured in the experiments)
* Report a meaningful summary of experimental results in the main paper

---

> ### Author Response · Authors · 2022-08-02
> **Response to Reviewer tvSG**
>
> We thank the reviewer for the thoughtful comments and suggestions. Our responses and clarifications are summarized below.
>
>
> **The motivation for using coresets (for the questions Q1 and Q2).**
>
> Actually constructing the design matrix indeed takes large time and space complexities. It is also worth emphasizing that for most of the previous research on relational algorithms [20,30,31,36,39], avoiding constructing the complete design matrix is always their basic motivation.
>
>
>  In the worst case, the design matrix can be as large as $N^s$; in practice, it is also usually much larger than $N$. For example, in our experiment the design matrix extracted by $\mathrm{Query\ 1}$ (Q1)  contains more than $4.0\times 10^8$ rows, while the largest table contains less than $10^7$ rows (i.e., $N<10^7$). Overall the design matrix is more than $50$ times larger than the input relational tables.
>  Our experimental results show that  our server has to run about three hours to extract the $\mathrm{Query\ 1}$'s design matrix (see Table 2 in our Supplements (sorry that there is a typo on the  title, where it should be ''The results of SVM and logistic regression on $\mathrm{Q1}$'')). Our proposed algorithm takes only a few minutes.
>
>
> **About the ''foreign key join'' (for the question Q2).**
>
>  We can consider a simple example with  three tables $T_1(a,b), T_2(a,c), T_3(a,d)$, where the attributes $a, c, d$ are the primary keys of table $T_1$, $T_2$ and $T_3$, respectively; the attribute $a$ is a foreign key in both table $T_2$ and $T_3$. In this example, $T_1\bowtie T_2$ and $T_1\bowtie T_3$ are all foreign key joins, but in the worst case, the design matrix of $T_1\bowtie T_2\bowtie T_3$ can be as large as $N^2$ (e.g., it can be as large as the Cartesian product of the attributes $b$ and $c$, if the attribute $a$ in $T_2$ and $T_3$ take the same value in all the rows). The size expansion can be even more significant if the number of tables is larger than three. In our experiments, the results show that the number of rows  of the design matrix  for $\mathrm{Query\ 1}$ is about $40N$.
>
>
>
> **''Q3 Suppose it would be feasible to efficiently materialize the join, what are the remaining challenges due to the relational data.........''**
>
> As mentioned in our first two responses, materializing the design matrix usually takes a large running time. Even if we can afford such a runtime, the matrix also takes a large space complexity in the memory. Thus, another benefit of our coreset technique on the relational tables is that it can save a large amount of space complexity, comparing with directly running the Gonzalez's algorithm on the design matrix.
>
>
> **''Q4 Which SQL queries are supported over the base tables?''**
>
> Thanks for this question, and we may need more explanation. Note that our goal is to extract the coreset from the given relational tables. Therefore, we only consider the queries that join the given tables together. It is out of the scope of this paper to consider other queries.
>
>
>
>
> **More details for ''pseudo-cube'' (for the question Q5).**
>
> To understand the geometry of ''pseudo-cube'', we can first imagine a special case: a $d$-dimensional hypercube with the center point $c=(c_1, c_2, \cdots, c_d)\in \mathbb{R}^d$ and the side length $2r$ ($r>0$). For any point $p=(p_1, p_2, \cdots, p_d) $ in the hypercube, we have $|c_i-p_i|<r, i\in\lbrace1,2,\ldots,d\rbrace$. This is a special case of the ''pseudo-cube'' of Definition 4 (line 240-250), where each subspace $\hat{D}_i$ is just a one-dimensional axis, and the number of subspaces $s=d$. For a general pseudo-cube, each subspace $\hat{D}_i$ can have a higher dimension and the condition ''$|p_i-x_i|\leq r$'' is replaced by ''$||\mathtt{Proj}\_{\hat{D}\_i}(c)-\mathtt{Proj}\_{\hat{D}\_i}(p)||\leq r$'' (i.e., $\mathtt{Proj}\_{\hat{D}\_i}(p)$ is inside the ball $\mathbb{B}(\mathtt{Proj}\_{\hat{D}\_i}(c), r)$). We can see that the pseudo-cube is a Cartesian product of a set of balls from the subspaces, which is not a standard cube (that is why we call it ''pseudo-cube'').

---

> > ### Comment · Reviewer_tvSG · 2022-08-05
> > **The work seems interesting and well-motivated, but there are some quality/clarity concerns**
> >
> > Thank you for the reply. The answer clears up questions 4 & 5 (question 3 was also answered in some other reply). With regards to questions 1 and 2:
> >
> > The multi-way foreign-key join definition in SIGMOD'99 (Definition 4.1 in https://doi.org/10.1145/304182.304207) is not met by the $T_1 \bowtie T_2 \bowtie T_3$ example and upon closer inspection does not seem to be met by the queries $Q1$ and $Q2$ in the supplement either. That explains the vast speed-ups reported in Table 1 and 2. The definition is not met, because there is no join ordering of these queries where all binary joins are two-way foreign key joins. Note that the design matrix size in case of multi-way foreign-key joins is always capped by $N$, because a two-way foreign key join cannot make the numbers of rows larger than in the largest involved table.
> >
> > It would therefore seem useful to distinguish at least three types of problem instances:
> >
> > - easy (foreign-key join queries as in SIGMOD'99 where design matrix size cannot exceed $N$)
> > - harder (other acyclic join queries such as $Q1$ & $Q2$ in the supplement or $QX$ in [52] where the design matrix can be much larger than $N$)
> > - hardest (cyclic join queries such as $QT$ in [52])
> >
> > ### Hardest:
> >
> > Cyclic joins are left as an open problem, which is perfectly fine.
> >
> > ### Harder:
> >
> > The harder problem instances are not as common (e.g., they are missing from popular database benchmarks like TPC-H), but they may still arise in practice (e.g., see suggestions in original review and prior work) and pose an interesting problem for coreset computation, because the design matrix may become prohibitively large. In this case the proposed methods are very interesting! The supplement already seems to feature experiments with harder problem instances (queries $Q1$ & $Q2$), but as far as I understand does not seem to report the design matrix runtime for query $Q3$ (is it an easier problem instance?).
> >
> > ### Easy:
> >
> > The easy ones are the most common and make it feasible to materialise the design matrix. It appears in that case one can retrieve the core set directly via Gonzalez's algorithm [24]. While the proposed method may still come with some small benefits (e.g., smaller memory footprint), it also comes with the disadvantage of only supporting naked joins while the naive method supports the whole SQL arsenal (W3). Currently, the manuscript not only omits the existence of such easy problem instances, but also seems to use them to motivate the work (cf. question 1). This raises some concerns that in the reviewing guidelines are phrased as follows:
> >
> > - "Quality: Is the submission technically sound? Are claims well supported (e.g., by theoretical analysis or experimental results)? [...]"
> > - "Clarity: [...] Does it adequately inform the reader?"
> >
> > Prior works such as [20] did not raise such concerns, because they made it quite clear that they focus on harder problem instances:
> >
> > - quote from [20] abstract: "When the data matrix needs to be obtained from a relational database via a feature extraction query, the computation cost *can* be prohibitive, as the data matrix *may* be (much) larger than the total input relation size."
> >
> > In contrast, in the manuscript it is simplified to a point where it may mislead readers:
> >
> > - quote from manuscript abstract: "In this paper, we consider the problem of coresets construction over relational data. Namely, the data is decoupled into several relational tables, and it *is* expensive to directly materialize the data matrix by joining the tables."
> >
> > ### Conclusion:
> >
> > It would really help to know if any changes will be made to the manuscript to address these quality/clarity concerns and what the nature of these changes would be.

---

> > > ### Author Response · Authors · 2022-08-06
> > > **Response to the "quality/clarity concerns"**
> > >
> > > We thank the reviewer for the clear explanation on these three types.
> > > Like most existing research on relational algorithm [20, 30, 36], we pay more attention to the "harder" instance.
> > > We will clarify it and modify our description in our paper.
> > >
> > > We agree that $\mathrm{Q3}$ is an easy instance which corresponds to the multi-way foreign key join. But our proposed algorithm still outperforms the $\mathrm{ORIGINAL}$ (construct the design matrix and run the learning algorithm on the whole design matrix) in terms of the runtime (see the left figure in Figure 1 in our supplement) and takes smaller space complexity.

---

> > > > ### Comment · Reviewer_tvSG · 2022-08-08
> > > > **Empirical Evaluation**
> > > >
> > > > Thank you for the response, this gives a much better idea about the problem setting!
> > > >
> > > > Supposing that it would not seem too difficult to clarify the problem setting, the manuscript may still need more work with regards to the empirical evaluation. Specifically, it is not clear why:
> > > >
> > > > - two very simple/natural baselines are missing, i.e., coresets via random sampling [43] and running Gonzalez's algorithm [24] on the full design matrix
> > > > - the mislabeling of the tables is easily fixable, but two out of the three queries are still quite similar (Q1/Q2) and tabular results are missing for the third query (Q3) ; perhaps it would be useful to also feature the public dataset Favorita from [3]?
> > > >
> > > > Otherwise, the proposed method seems to have some merit (very interesting to achieve core set size dependent on doubling dimension rather than data dimensionality) and offer some reasonable solutions (approximating core set weights via random sampling) to the pseudo-cube overlap problem that arises when dealing with multiple relational tables. Thus, it would be very useful to know if there is some commitment to extend the empirical evaluation with regards to the two raised points (and adding some meaningful discussion of results in the main paper).

---

> > > > > ### Author Response · Authors · 2022-08-08
> > > > > **Response to "Empirical Evaluation"**
> > > > >
> > > > > We appreciate the helpful suggestions. We did some preliminary experiments on uniform random sampling [43] before, and the results show that the performance of uniform sampling is relatively poor. Also as mentioned in [43], the required coreset size  by uniform sampling can be as large as $\Theta\left( \sqrt{n} \cdot d \right)$. But for completeness, we will add the two baselines and the dataset from [3] with more discussions to our paper as suggested by the reviewer.

---

> > > > > > ### Comment · Reviewer_tvSG · 2022-08-08
> > > > > > **Thank you**
> > > > > >
> > > > > > Thank you for the responses, that makes it a lot clearer.

---

### Official Review · Reviewer_SwdD · 2022-07-11

**Rating:** 7
**Confidence:** 5
**Soundness:** 3 good
**Presentation:** 4 excellent
**Contribution:** 3 good

**Summary:**

This paper claims that when given a relational database that is normalized, we don't need to explicitly create the design matrix by denormalizing the data and then sample them with the coresets. In other world when the data ialredy given compressed as a normalized database we can reduce them further with coresets without having to create a huge temporary dataset. If we know the application of coresets such as logistic regression and SVMs we use the recent results of the FAQ method to push the computations before the joins.

**Questions:**

Given the impressive speedups where we can think of SVMs and logistic regression as fast block operators, what else can we do?

**Ethics Review Area:**

["I don’t know"]

**Limitations:**

Although the results on SVMs and logistic regression are impressive, the truth is that these methods are already fast and streaming methods like vopal wabbit are already popular. It would be interesting to see how this technique would apply to deep learning. It is true that after the first layer the fully connected layers are not relational anymore. There is a special type of neural networks like the columns neural networks https://arxiv.org/pdf/1904.06950.pdf that preserve the relational structure.

**Strengths And Weaknesses:**

The paper presents the use of some fundamental database algorithms such as FAQ in a classic machine learning problem the SVMs.  The assumption that the data can be relational is valid and true in real-world scenarios. Relational data have been neglected by the ML community despite their popularity. The paper provides solid theoretical explanation of the method with complexity bounds.
The benchmarks show impressive speedups, but I would have preferred more extensive experiments.
I was also not very happy with the lack of references on kd-trees and other related ones. The method they propose looks like a relational kd-tree, or there seems to be a connection. I think it is worth exploring.

---

> ### Author Response · Authors · 2022-08-02
> **Response to Reviewer SwdD**
>
> We thank the reviewer for the thoughtful comments and suggestions. Our responses and clarifications are summarized below.
>
>
> **''....the lack of references on kd-trees and other related ones...''**
>
>
> Thanks and we will add the references on kd-tree and the related ones. But it is worth emphasizing that our aggregation tree and kd-tree are quite different structures, and they are used for different applications (our method is for compressing data, and kd-tree is for nearest neighbor search). We will add more details to explain this point in our paper.
>
> **''Given the impressive speedups where we can think of SVMs and logistic regression as fast block operators, what else can we do?'', ''It would be interesting to see how this technique would apply to deep learning. ''**
>
>
> Thanks for posing this question. According to our Theorem 1,  our coreset technique actually can be applied to any machine learning objective that satisfies Assumption 1. We agree that it is also interesting to consider deep learning, though it may be much more challenging than other machine learning tasks. We will try to extend our technique to deep learning as the future work.

---

> > ### Comment · Reviewer_SwdD · 2022-08-08
> > **Response**
> >
> > I do understand the kd-trees are more applicable to numerical values, but the nodes can be used as core sets in many algorithms. They don't take advantage of functional dependencies but they could be extended that way. In any event, I thought that a reference or a connection to kd-trees could help the reader anchor on a well-known data structure

---

> > > ### Author Response · Authors · 2022-08-09
> > > **Thanks for the suggestion**
> > >
> > > Sure, we will add the references and some discussions on kd-tree. Thanks for the suggestion.

---

### Meta-Review · Area_Chair_n84w · 2022-08-21

**Recommendation:** Accept
**Confidence:** Certain

**Metareview:**

The paper considers the problem of coresets construction over relational data, where the data is decoupled into several relational tables, and it is expensive to directly materialize the data matrix by joining the tables. The paper addresses this problem by proposing ``aggregation tree with pseudo-cube'' that builds a coreset from bottom to up. The problem is interesting, the proposed solution is valuable to the community, and the reviewers found the rebuttal very helpful. However, after extensive discussion the reviewers emphasized that the paper is **acceptable if** the authors add the following two experiments they promised in the rebuttal: (1) two natural baselines, and (2) at least one more query from another dataset.

**Award:**

No

---

### Decision · Program_Chairs · 2022-09-14

Accept